# Taxifolin as a Therapeutic Potential for Weight Loss: A Retrospective Longitudinal Study

**DOI:** 10.3390/nu17040706

**Published:** 2025-02-16

**Authors:** Yorito Hattori, Yuriko Nakaoku, Soshiro Ogata, Satoshi Saito, Kunihiro Nishimura, Masafumi Ihara

**Affiliations:** 1Department of Neurology, National Cerebral and Cardiovascular Center, 6-1 Kishibe-shimmachi, Suita 564-8565, Osaka, Japan; yoh2019@ncvc.go.jp (Y.H.);; 2Department of Preemptive Medicine for Dementia, National Cerebral and Cardiovascular Center, 6-1 Kishibe-shimmachi, Suita 564-8565, Osaka, Japan; 3Department of Preventive Medicine and Epidemiology, National Cerebral and Cardiovascular Center, Suita 564-8565, Osaka, Japan

**Keywords:** taxifolin, weight loss, high-density lipoprotein cholesterol, brown adipose tissue

## Abstract

Background/Objectives: The current approach to obesity care, which primarily focuses on weight loss, is often insufficient because of the challenges in maintaining long-term results. Therefore, novel, safe, and sustainable medications for obesity are highly anticipated. Taxifolin, a natural bioactive flavonoid, was found to exert pleiotropic protective effects against various diseases. Our experimental *in vivo* and *in vitro* studies revealed that taxifolin administration contributes to weight loss. Accordingly, we hypothesized that long-term oral intake of taxifolin was clinically associated with weight loss. Methods: A retrospective longitudinal study was conducted on participants who consistently monitored their body weight during routine clinic visits between January 2021 and July 2021. Body weight changes of the patients who received 300 mg/day of taxifolin were compared with those of patients who did not receive taxifolin. Results: The study enrolled a total of 62 patients: 36 received taxifolin and 26 did not receive taxifolin. Long-term intake of taxifolin showed greater weight loss than those not receiving taxifolin over a mean follow-up of 176.1 and 177.7 days, respectively (−1.6 vs. −0.3 kg; *p* = 0.026). Furthermore, long-term taxifolin intake was an independent predictor of increased weight loss (adjusted β [mean difference] −0.14, 95% confidence interval [−2.69, −0.18], *p* = 0.026). No adverse events were observed. Conclusions: Long-term daily oral intake of taxifolin may safely and sustainably prevent or manage obesity.

## 1. Introduction

Obesity is an escalating public health issue and ranks as the most prevalent chronic disease globally, affecting approximately 650 million adults [1]. Excess adiposity and its comorbidities such as cardiovascular disease, type 2 diabetes mellitus (DM), cancer, hypertension, and musculoskeletal disorders, impose a substantial economic burden and are key contributors to global morbidity and mortality rates. Treatments leading to considerable weight loss may improve the outcomes for individuals with obesity [1]. Diet and physical activity have been the primary interventions for weight reduction over the past decade [2]. However, maintaining weight loss is challenging, and the availability of evidence-based treatment is limited [3].

Taxifolin (TAX; known as 3,5,7,3’,4’-pentahydroxy flavanone or dihydroquercetin) is a natural bioactive flavonoid [4]. TAX has attracted increased attention as a potential treatment for DM [5], cardiovascular diseases [6], and early Alzheimer’s disease [7] through positive mechanisms, including anti-inflammatory, antioxidant, antiapoptotic, and mitochondrial protective effects [8]. TAX also mitigates obesity development by increasing the mRNA levels of uncoupling protein-1 and fibroblast growth factor 21 (FGF21) in brown adipose tissue and high body temperatures in our *in vivo* and *in vitro* study [9] and activating the phosphatidylinositol 3-kinase/protein kinase B signaling pathway in rats [10]. Therefore, we hypothesized that long-term oral intake of TAX could clinically contribute to weight loss.

## 2. Materials and Methods

### 2.1. Study Design

This single-center retrospective longitudinal study was conducted at the National Cerebral and Cardiovascular Center (NCVC). The study was approved by the Research Ethics Committee of the NCVC (Approval no. R24045; date of approval, 8 October 2024) and conducted in accordance with the standards of the Declaration of Helsinki. Opt-out consent was implemented, indicating that participants were automatically included in the study unless they expressed a desire to withdraw. The electronic medical charts of outpatients, obtained from January 2021 to July 2021, were screened to identify patients who (1) had regular measurements of body weight during clinic visits with an interval of 180 ± 100 days and (2) provided written informed consent for NCVC Biobank. Among patients who fulfilled the criteria, some voluntarily purchased TAX (300 mg/day) after a clinical research coordinator provided the following information to them and/or their families for >30 min, independent of the attending physicians: TAX inhibits lipid accumulation [11] and decreases body weight [10] in preclinical studies; however, its clinical effectiveness is unclear. TAX tablets manufactured by Towa Pharmaceutical Co., Ltd. (Kadoma, Osaka, Japan), are commercially available. Clinical data, including age, sex, vascular risk factors such as hypertension (defined by a history of antihypertensive drug use), DM (defined as hemoglobin A1c ≥ 6.5%, or a history of treatment for DM), dyslipidemia (defined as low-density lipoprotein cholesterol level ≥ 140 mg/dL, triglyceride level ≥ 175 mg/dL, high-density lipoprotein cholesterol [HDL-C] level < 40 mg/dL, or a history of treatment for dyslipidemia), and drugs, were obtained.

### 2.2. Statistical Analysis

Patient characteristic data are summarized as mean ± standard deviation for continuous variables and as frequencies and percentages for categorical variables. Statistical differences for continuous and categorical variables were assessed using Student’s *t*-test and the chi-square test, respectively. Temporal changes in parameters, such as body weight and blood test parameters, were calculated by subtraction. To assess the difference in temporal changes in body weight between the TAX and non-TAX groups, multivariable linear regression analysis was performed after adjustment for age, sex, hypertension, DM, dyslipidemia, antihypertensive drugs, anti-DM drugs, statins, and baseline body weight. To assess the difference in temporal changes in the body mass index (BMI) between the TAX and non-TAX groups, multivariable linear regression analysis was performed after adjustment for age, sex, hypertension, DM, dyslipidemia, antihypertensive drugs, anti-DM drugs, statins, and baseline BMI. To investigate the association between TAX intake and temporal changes in blood HDL-C levels, multivariable linear regression analysis was performed after adjusting for age, sex, dyslipidemia, statins, and baseline body weight. Pearson’s correlation analysis between temporal changes in body weight and blood HDL-C levels was also performed. All reported *p* values were two-sided, and *p* values <0.05 were considered significant. All analyses were performed using IBM SPSS software version 29 (IBM Corp., Armonk, NY, USA) and GraphPad PRISM 9 (GraphPad Software, Boston, MA, USA).

## 3. Results

### 3.1. Baseline Characteristics

This study enrolled 62 patients who had regular measurements of their body weight during clinic visits. Among the 62 patients, 26 did not take TAX (non-TAX group), and 36 started to take TAX (TAX group). The mean age (77.5 vs. 76.8 years; *p* = 0.70), proportion of male patients (16 [61.5%] vs. 16 [44.4%]; *p* = 0.18), mean body weight (58.7 vs. 60.1 kg; *p* = 0.61), mean BMI (23.5 vs. 23.6 kg/m^2^; *p* = 0.88), and observation periods (177.7 vs. 176.1 days; *p* = 0.81) were comparable between the two groups. The frequency of comorbidities, such as hypertension, DM, and dyslipidemia; antihypertensive and anti-DM drugs; and statins was not different between the two groups (Table 1).

### 3.2. TAX-Intake Was Associated with Significant Weight Loss and Decrease in BMI

First, body weight changes were compared between the TAX and non-TAX groups. The TAX group showed greater weight loss than the non-TAX group (−0.3 vs. −1.6 kg; *p* = 0.026) (Figure 1A). To determine if long-term TAX intake was an independent predictor of increased weight loss, linear regression analyses were conducted. Long-term TAX intake was an independent predictor of greater weight loss (model 1, β [mean difference in Δ body weight] −1.52, 95% confidence interval [CI] [−2.85, −0.19], *p* = 0.026; model 2, β [mean difference in Δ body weight] −1.52, 95% CI [−2.69, −0.18], *p* = 0.026; model 3, β [mean difference in Δ body weight] −1.78, 95% CI [−3.15, −0.40], *p* = 0.013) (Figure 1C).

Similarly, BMI changes were compared between the TAX and non-TAX groups. The TAX group tended to exhibit a greater decrease in BMI than the non-TAX group (−0.11 vs. −0.61; *p* = 0.051) (Figure 1B). To determine whether long-term TAX intake was an independent predictor of a greater decrease in BMI, linear regression analyses were performed. Long-term TAX intake was identified as an independent predictor of a greater decrease in BMI across all models (model 1, β [mean difference in Δ BMI] −0.55, 95% CI [−1.09, −0.020], *p* = 0.042; model 2, β [mean difference in Δ BMI] −0.57, 95% CI [−1.11, −0.025], *p* = 0.041; model 3, β [mean difference in Δ body weight] −0.57, 95% CI [−1.12, −0.024], *p* = 0.041) (Figure 1D). No adverse events were observed.

### 3.3. TAX-Associated Weight Loss Was Correlated with Increase in Blood HDL-C Level

To investigate the mechanisms of TAX-associated weight loss, changes in blood parameter were compared between the two groups. Long-term TAX intake significantly suppressed the reduction in blood HDL-C levels (−6.17 vs. −0.86 mg/dL, *p* = 0.008) (Figure 2C). The other blood metabolic markers were comparable (Figure 2A,B,D,E,F).

Multivariable linear regression analysis was performed to investigate whether long-term TAX intake was associated with an increase in blood HDL-C levels because of the significant difference in blood HDL-C levels only between the non-TAX and TAX groups. The multivariable linear regression analysis revealed that long-term TAX intake was significantly associated with an increase in blood HDL-C levels (model 1, β [mean difference in Δ blood HDL-C level] 5.04, 95% CI [1.10, 8.99], *p* = 0.013; model 2, β [mean difference in Δ blood HDL-C level] 4.95, 95% CI [0.96, 8.93], *p* = 0.016; model 3, β [mean difference in Δ blood HDL-C level] 4.89, 95% CI [0.81, 8.96], *p* = 0.020) (Figure 3).

In the TAX group, increased blood HDL-C levels were significantly correlated with weight loss (Figure 4A). In contrast, the non-TAX group did not exhibit a correlation between weight loss and an increase in blood HDL-C levels (*r* = 0.002, *p* = 0.99) (Figure 4B). Therefore, blood HDL-C levels should be preserved to help achieve weight loss.

## 4. Discussion

This retrospective longitudinal study revealed a significant association between long-term TAX intake and weight loss. This result was consistent with previous experimental studies [9,10]. In only the TAX group, an increase in blood HDL-C levels was significantly correlated and associated with weight loss.

Several theories have been proposed for its underlying mechanism involving the brown adipose tissue and HDL-C. In our experimental study, TAX enhanced the function of brown adipose tissue by promoting the production of batokines, including uncoupling protein 1 (UCP1), which is related to nonshivering thermogenesis, and FGF21 [9]. Brown adipose tissue, which is involved in diet-induced thermogenesis, contributes to overall energy expenditure and exerts antiobesity effects [12]. Thermogenic adipocytes express high levels of UCP1, which dissipates energy as heat by uncoupling the mitochondrial proton gradient from mitochondrial respiration. Substantial evidence supports the central role of UCP1 in brown adipose tissue thermogenesis and systemic energy homeostasis [13]. FGF21 also promotes thermogenic activity and ketogenesis [14]. Nutritional ketosis was found to improve the metabolic condition and aid in weight management [15]. Thus, TAX may play an important role in weight loss by increasing the thermogenetic activity of brown adipose tissue.

The relationship between TAX intake and high blood HDL-C levels involves a crosstalk between brown tissue adipocytes or batokines and HDL-C. Subcutaneous injection of a human FGF21 analog dose-dependently increases blood HDL-C levels and results in weight loss in individuals aged 18–75 years with a body mass index of 25–40 kg/m^2^ [16]. Furthermore, the increased activity of brown adipose tissue, as evaluated by ^18^F-fluoro-deoxy-glucose positron emission tomography/computed tomography, was significantly associated with high HDL-C levels in the blood [17]. Thus, the TAX-associated increase in blood HDL-C levels could reflect brown adipose tissue activation or thermogenesis, which can lead to weight loss. Unfortunately, this study lacks sufficient power to evaluate the relationship between TAX intake and various metabolic markers because the activated brown adipose tissue and batokines have been linked to decreased blood glucose and improved lipid profiles, including triglycerides and low-density lipoprotein cholesterol [16,17].

This retrospective study had several limitations. First, patients receiving the placebo control were not compared with those receiving TAX, which indicates that only patients with a higher level of health consciousness voluntarily purchased TAX. Second, the sample size was small owing to the exploratory nature of this study, which may lead to issues, such as limited generalizability, low statistical power, exaggerated effect size, and reduced variability. Third, data on food consumption were not collected because of the retrospective and exploratory nature of the study. Thus, future randomized, double-blind, placebo-controlled trials incorporating larger sample sizes are needed. Despite these limitations, the results indicate that long-term oral TAX intake may promote weight loss in older individuals.

## 5. Conclusions

Long-term oral intake of TAX yielded significant weight loss, possibly via the activation of the brown adipose tissue, without any adverse events. As weight loss and maintenance are still challenging problems, the development of safe and sustainable preventive medications is eagerly awaited. Our results highlight the need for a randomized, double-blind, placebo-controlled trial to analyze the effects of TAX on weight loss.

## Figures and Tables

**Figure 1 nutrients-17-00706-f001:**
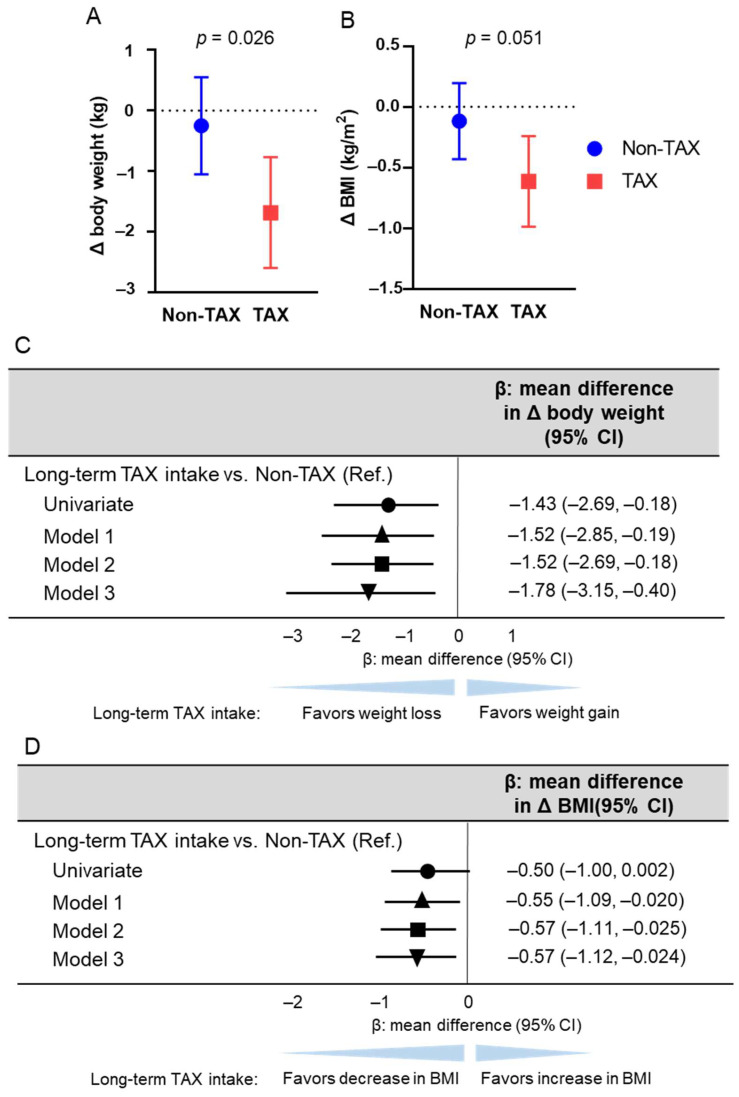
Association between long-term taxifolin (TAX) intake and weight loss and decrease in body mass index (BMI). The means of body weight (**A**) and BMI (**B**) changes and error bars in the non-TAX and TAX groups. Error bars indicate 95% confidence interval (CI). (**C**) Linear regression analysis estimating long-term TAX intake associated with weight loss. The multivariable models were adjusted for models 1, 2, and 3. Model 1 was adjusted for age, sex, hypertension, diabetes mellitus, and dyslipidemia; model 2 was further adjusted for antihypertensive drug use, antidiabetic drug use, and statin use; and model 3 was further adjusted for baseline body weight. (**D**) Linear regression analysis estimating long-term TAX intake associated with the decrease in BMI. The multivariable models were adjusted for models 1, 2, and 3. Model 1 was adjusted for age, sex, hypertension, diabetes mellitus, and dyslipidemia; model 2 was further adjusted for antihypertensive drug use, antidiabetic drug use, and statin use; and model 3 was further adjusted for baseline BMI. β values are indicated with the circle (univariate), the triangle (Model 1), the square (Model 2), and the inverted triangle (Model 3).

**Figure 2 nutrients-17-00706-f002:**
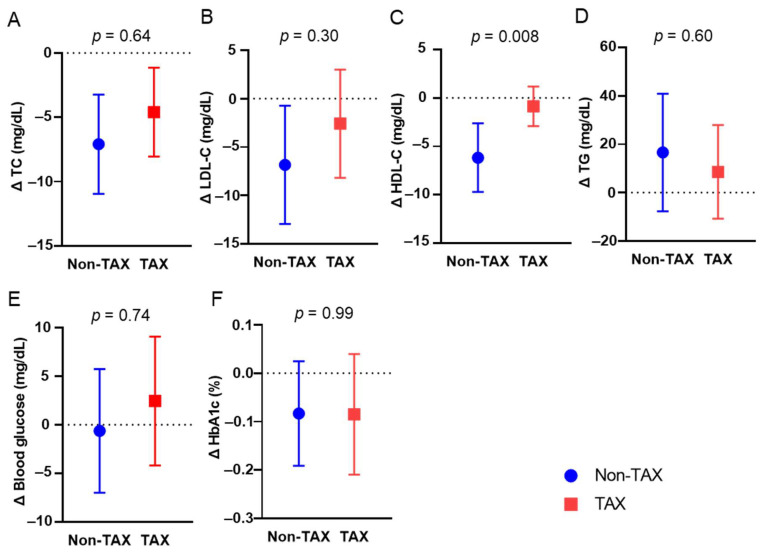
Comparison of blood metabolic markers between the non-taxifolin (TAX) and TAX groups. The means and error bars of total cholesterol (TC) (**A**), blood low-density lipoprotein cholesterol (LDL-C) (**B**), high-density lipoprotein cholesterol (HDL-C) (**C**), triglyceride (TG) (**D**), routine blood glucose (**E**), and HbA1c (**F**) levels in the non-TAX and TAX groups. Error bars indicate 95% confidence interval (CI).

**Figure 3 nutrients-17-00706-f003:**
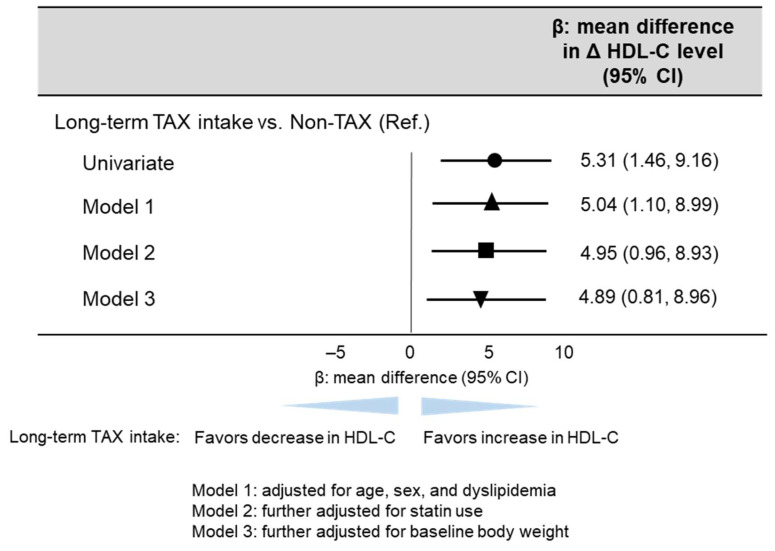
Linear regression analysis estimating long-term TAX intake associated with blood HDL-C change. The multivariable models were adjusted for models 1, 2, and 3 as follows: model 1: adjusted for age, sex, and dyslipidemia; model 2: further adjusted for statin use; and model 3: further adjusted for baseline body weight. β values are indicated with the circle (univariate), the triangle (Model 1), the square (Model 2), and the inverted triangle (Model 3).

**Figure 4 nutrients-17-00706-f004:**
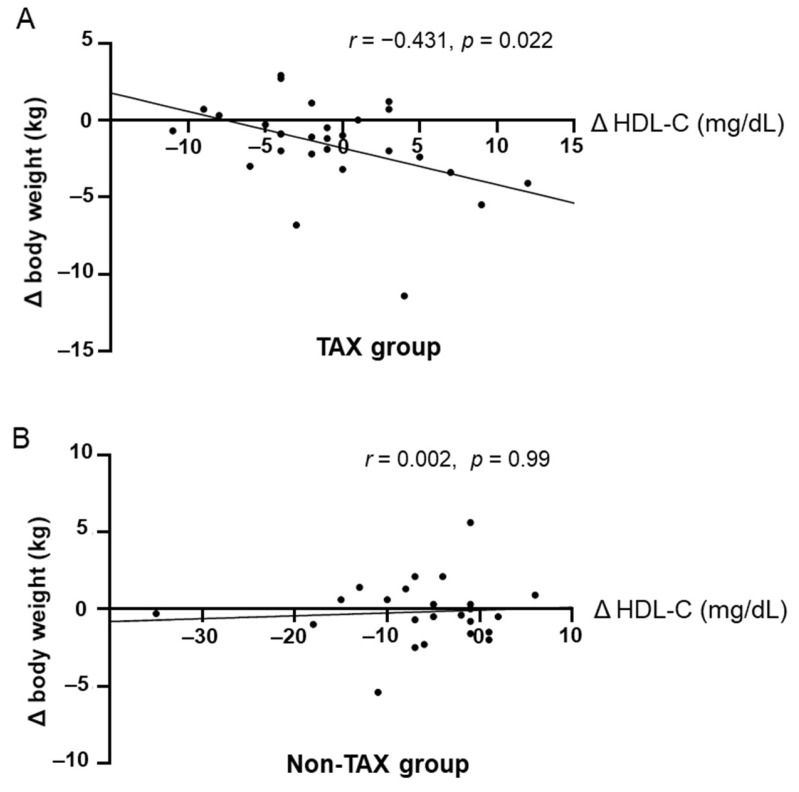
Correlations between increase in blood high-density lipoprotein cholesterol (HDL-C) levels and weight loss. Pearson’s correlation analyses between blood HDL-C levels and body weight change in the taxifolin (TAX) group (**A**) and non-TAX group (**B**).

**Table 1 nutrients-17-00706-t001:** Baseline characteristics of the patients.

	Non-TAX Group	TAX Group	*p* Value
Number	26	36	—
Age	77.5 ± 5.7	76.8 ± 7.2	0.70
Sex, male	16 (61.5%)	16 (44.4%)	0.18
Body weight [kg]	58.7 ± 11.0	60.1 ± 10.1	0.61
Body mass index [kg/m^2^]	23.5 ± 3.3	23.6 ± 3.2	0.88
Hypertension	18 (69.2%)	29 (80.6%)	0.30
Diabetes mellitus	8 (30.8%)	5 (13.9%)	0.11
Dyslipidemia	19 (73.1%)	25 (69.4%)	0.76
Antihypertensive drug use	18 (69.2%)	29 (80.6%)	0.30
Statin use	17 (65.4%)	22 (61.1%)	0.73
Antidiabetic drug use	5 (19.2%)	4 (11.1%)	0.37
Blood glucose [mg/dL]	120.7 ± 35.7	108.9 ± 26.7	0.24
Hemoglobin A1c [%]	6.2 ± 0.9	6.0 ± 0.6	0.34
Total cholesterol [mg/dL]	179.1 ± 34.4	183.4 ± 35.6	0.65
Triglyceride [mg/dL]	121.5 ± 49.2	122.6 ± 70.7	0.95
LDL-C [mg/dL]	98.4 ± 28.9	97.8 ± 26.1	0.93
HDL-C [mg/dL]	58.6 ± 14.7	61.5 ± 14.5	0.47
Observational period [days]	177.7 ± 18.5	176.1 ± 29.1	0.81

Data are presented as the mean ± standard deviation or number (%). Abbreviations: TAX, taxifolin; LDL-C, low-density lipoprotein cholesterol; HDL-C, high-density lipoprotein cholesterol.

## Data Availability

The data supporting the findings of this study are available on request from the corresponding author. The data are not publicly available due to privacy or ethical restrictions.

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
