# Peer review of "Taxifolin as a Therapeutic Potential for Weight Loss: A Retrospective Longitudinal Study"

_nutrients, 2025, doi:10.3390/nu17040706_

Round 1
Reviewer 1 Report
Comments and Suggestions for Authors
The manuscript presents concisely results of an experiment on the effect of a long-term (18- or 19-month) supplementation with taxifolin on the weight of elderly patients during. Although the size of the experimental group was small, the results show unequivocally to the weight loss in persons receiving taxifolin, in contrast to the control group. These results are interesting and should be tested on a larger group. Possible mechanism of the effect is suggested, basing on previous studies.
The limitations of the study, including the small size of the group and lack of placebo are properly discussed.
Remarks:
Reporting BMI would be desirable (if data are available).
Ded the supplementation affect the appetite/food consumption?
The dose administered in reported only in Abstract; it would be desirable to report details of supplementation under Materials and Methods.
Lines 81/82: “DM, dyslipidemia, antihypertensive drugs, anti-DM drugs, statins, and baseline 81 body weight.”, apparently, part of the sentence was cut off. Please correct.
Lines 196-198: Please remove the first sentence, which is an instruction.
References: Volume numbers in italics, please
Author Response
Reviewer #1.
The manuscript presents concisely results of an experiment on the effect of a long-term (18- or 19-month) supplementation with taxifolin on the weight of elderly patients during. Although the size of the experimental group was small, the results show unequivocally to the weight loss in persons receiving taxifolin, in contrast to the control group. These results are interesting and should be tested on a larger group. Possible mechanism of the effect is suggested, basing on previous studies.
The limitations of the study, including the small size of the group and lack of placebo are properly discussed.
Remarks:
Reporting BMI would be desirable (if data are available).
RESPONSE: Thank you for the suggestion. Height data were collected from medical charts, and BMI was calculated. The baseline BMI was comparable between the non-TAX and TAX groups (p = 0.88). The result was added in Table 1. Furthermore, BMI changes were compared between the two groups. The TAX group demonstrated a trend toward a greater decrease in BMI compared with the non-TAX group (−0.11 vs. −0.61; p = 0.051) (new Figure 1B). After adjusting for model 3, long-term TAX intake was identified as an independent predictor of a greater decrease in BMI (β [mean difference in Δ BMI] −0.57, 95% CI [−1.12, −0.024], p = 0.041) (new Figure 1D).
(Lines 82–85)
To assess the difference in temporal changes in body mass index (BMI) between the TAX and non-TAX groups, multivariable linear regression analysis was performed after adjustment for age, sex, hypertension, DM, dyslipidemia, antihypertensive drugs, anti-DM drugs, statins, and baseline BMI.
(Lines 115–123)
Similarly, BMI changes were compared between the TAX and non-TAX groups. The TAX group tended to exhibit a greater decrease in BMI than the non-TAX group (−0.11 vs. −0.61; p = 0.051) (Figure 1B). To determine whether long-term TAX intake was an independent predictor of a greater decrease in BMI, linear regression analyses were performed. Long-term TAX intake was identified as an independent predictor of a greater decrease in BMI across all models (model 1, β [mean difference in Δ BMI] −0.55, 95% CI [−1.09, −0.020], p = 0.042; model 2, β [mean difference in Δ BMI] −0.57, 95% CI [−1.11, −0.025], p = 0.041; model 3, β [mean difference in Δ body weight] −0.57, 95% CI [−1.12, −0.024], p = 0.041) (Figure 1D).
(Lines 132–137)
(D) Linear regression analysis estimating long-term TAX intake associated with the decrease in BMI. The multivariable models were adjusted for models 1, 2, and 3. Model 1 was adjusted for age, sex, hypertension, diabetes mellitus, and dyslipidemia; model 2 was further adjusted for antihypertensive drug use, antidiabetic drug use, and statin use; and model 3 was further adjusted for baseline BMI.
Ded the supplementation affect the appetite/food consumption?
RESPONSE: Thank you for raising this important point. Unfortunately, we did not collect data on food consumption because of the retrospective and exploratory nature of the study. Therefore, we have added this point as a study limitation. However, even some placebo-controlled randomized clinical trials investigating weight changes did not consider the amount of food consumption (Chandradasa, et al. Asian J Psychiatr. 2022; Pratley, et al. Lancet. 2019; Trouillot, et al. Am J Gastroenterol. 2001; Guo, et al. Diabetes Care. 2017; Hollander, et al. Diabetes Care. 2013).
(Lines 208–209)
Third, data on food consumption were not collected because of the retrospective and exploratory nature of the study.
The dose administered in reported only in Abstract; it would be desirable to report details of supplementation under Materials and Methods.
RESPONSE: Thank you for raising this point. We have added the dose in 2.1 Study design.
Lines 81/82: “DM, dyslipidemia, antihypertensive drugs, anti-DM drugs, statins, and baseline 81 body weight.”, apparently, part of the sentence was cut off. Please correct.
RESPONSE: Thank you for the correction. “DM, dyslipidemia, antihypertensive drugs, anti-DM drugs, statins, and baseline body weight” should have been directly followed by the immediately preceding sentence “multivariable linear regression analysis was performed after adjustment for age, sex, hypertension.” Accordingly, we have combined the two parts as follows:
(Lines 79–81)
…multivariable linear regression analysis was performed after adjustment for age, sex, hypertension, DM, dyslipidemia, antihypertensive drugs, anti-DM drugs, statins, and baseline body weight.
Lines 196-198: Please remove the first sentence, which is an instruction.
RESPONSE: Thank you for the correction. We have removed the first sentence.
References: Volume numbers in italics, please
RESPONSE: Thank you for raising this point. We have replaced the normal fonts with italics.
Reviewer 2 Report
Comments and Suggestions for Authors
In this retrospective longitudinal study authors hypothesized that long-term oral intake of taxifolin was clinically associated with weight loss.
1) Authors should add the potential biological mechanism that correlates HDL-C increase and weight loss.
2) In the discussion section, authors add the several theories proposed to understand the underlying mechanism involving the brown adipose tissue and HDL-C.
3) How the study limitations could impact the interpretation of the results?
4) As mentioned by the authors, this retrospective study had several limitations. In particular, they did not evaluate the relationship between TAX intake and several metabolic markers. In my opinion, this relationship should be integrated to complete the study.
Author Response
Reviewer #2.
In this retrospective longitudinal study authors hypothesized that long-term oral intake of taxifolin was clinically associated with weight loss.
1) Authors should add the potential biological mechanism that correlates HDL-C increase and weight loss.
RESPONSE: That is one of the important points. We already indicated the potential biological mechanisms in the Discussion (Lines 177–198). In our previous experimental study, we revealed that taxifolin enhanced the function of the brown adipose tissue by promoting the production of batokines, such as uncoupling protein 1, which is related to nonshivering thermogenesis, and FGF21. The batokines increase blood HDL-C levels. Thus, a taxifolin-associated increase in blood HDL-C levels could reflect brown adipose tissue activated by taxifolin, or thermogenesis, leading to weight loss. We have added the explanation as follows:
(Lines 196–198)
Thus, the TAX-associated increase in blood HDL-C levels could reflect brown adipose tissue activation or thermogenesis, which can lead to weight loss.
2) In the discussion section, authors add the several theories proposed to understand the underlying mechanism involving the brown adipose tissue and HDL-C.
RESPONSE: The underlying mechanism involving the brown adipose tissue and HDL-C has received significant attention. As stated in the Discussion (Lines 177–298), our experimental study revealed that taxifolin enhanced the function of brown adipose tissue by promoting the production of batokines, including UCP1 and FGF21. Furthermore, the administration of a human FGF21 analog increases blood HDL-C levels in old individuals. The increased activity of brown adipose tissue, as evaluated by 18F-fluoro-deoxy-glucose positron emission tomography/computed tomography, was significantly associated with high HDL-C levels in the blood. Thus, the TAX-associated increase in blood HDL-C levels in this study could reflect brown adipose tissue activation or thermogenesis.
3) How the study limitations could impact the interpretation of the results?
RESPONSE: Thank you for the question. First, the study lacked a placebo group, indicating that only patients with a higher level of health consciousness voluntarily purchased taxifolin. Second, the small sample size may lead to issues, such as limited generalizability, low statistical power, exaggerated effect size, and reduced variability. A future clinical trial should address these limitations by including a placebo group and enrolling a larger sample size. We have added the implications of the study limitations in the limitation paragraph, as shown below:
(Lines 203–208)
First, patients receiving placebo control were not compared with those receiving TAX, which indicates that only patients with a higher level of health consciousness voluntarily purchased TAX. Second, the sample size was small owing to the exploratory nature of this study, which may lead to issues, such as limited generalizability, low statistical power, exaggerated effect size, and reduced variability.
4) As mentioned by the authors, this retrospective study had several limitations. In particular, they did not evaluate the relationship between TAX intake and several metabolic markers. In my opinion, this relationship should be integrated to complete the study.
RESPONSE: Thank you for the valuable suggestion. We have added investigations of the associations of long-term TAX intake with BMI and total cholesterol, and routine blood glucose levels, in addition to associations of long-term TAX intake with LDL cholesterol, HDL cholesterol, triglyceride levels, and HbA1c, as shown in Table 1 and Figures 1–3.
Regarding the BMI, the height data were collected from medical charts, and BMI was calculated. Baseline BMI was comparable between the non-TAX and TAX groups (p = 0.88). The result was added in Table 1. Furthermore, BMI changes were compared between the two groups. The TAX group demonstrated a trend toward a greater decrease in BMI compared with the non-TAX group (−0.11 vs. −0.61; p = 0.051) (new Figure 1B). After adjusting for model 3, long-term TAX intake was identified as an independent predictor of a greater decrease in BMI (β [mean difference in Δ BMI] −0.57, 95% CI [−1.12, −0.024], p = 0.041) (new Figure 1D).
(Lines 82–85)
To assess the difference in temporal changes in the body mass index (BMI) between the TAX and non-TAX groups, multivariable linear regression analysis was performed after adjustment for age, sex, hypertension, DM, dyslipidemia, antihypertensive drugs, anti-DM drugs, statins, and baseline BMI.
(Lines 115–123)
Similarly, BMI changes were compared between the TAX and non-TAX groups. The TAX group tended to exhibit a greater decrease in BMI than the non-TAX group (−0.11 vs. −0.61; p = 0.051) (Figure 1B). To determine whether long-term TAX intake was an independent predictor of a greater decrease in BMI, linear regression analyses were performed. Long-term TAX intake was identified as an independent predictor of a greater decrease in BMI across all models (model 1, β [mean difference in Δ BMI] −0.55, 95% CI [−1.09, −0.020], p = 0.042; model 2, β [mean difference in Δ BMI] −0.57, 95% CI [−1.11, −0.025], p = 0.041; model 3, β [mean difference in Δ body weight] −0.57, 95% CI [−1.12, −0.024], p = 0.041) (Figure 1D).
(Lines 132–137)
(D) Linear regression analysis estimating long-term TAX intake associated with the decrease in BMI. The multivariable models were adjusted for models 1, 2, and 3. Model 1 was adjusted for age, sex, hypertension, diabetes mellitus, and dyslipidemia; model 2 was further adjusted for antihypertensive drug use, antidiabetic drug use, and statin use; and model 3 was further adjusted for baseline BMI.
Meanwhile, changes in blood total cholesterol and routine blood glucose levels were not associated with long-term intake of TAX (new Figures 2A and E).